# Feeding Preferences of Giant Pandas May Reflect the Detection of Specific Volatiles and Bitter-Tasting Metabolites in Bamboo Leaves as Markers of Nutritional Status

**DOI:** 10.3390/plants14243856

**Published:** 2025-12-18

**Authors:** Chao Bai, Yuyan You, Yanhui Liu, Haihong Xu, Yuanyuan Zhang, Guoyu Shan, Ali Wu, Liu Yang, Nan Ding, Yan Lu, Ting Jia, Yanping Lu, Yipeng Cong, Chenglin Zhang, Xuefeng Liu

**Affiliations:** 1Beijing Key Laboratory of Captive Wildlife Technologies, Beijing Zoo, Beijing 100044, China; baichao37@126.com (C.B.); youyy351@163.com (Y.Y.);; 2Beijing Zoo, Beijing 100044, China; 3Shanxi Changqing National Nature Reserve Administration, Hanzhong 723000, China

**Keywords:** feeding preferences of giant pandas, metabolites, volatiles, bamboo

## Abstract

Giant pandas feed preferentially on bamboo but choose different species and organs depending on factors such as the altitude and season, suggesting preferential selection according to their nutritional requirements. However, the mechanism of selection is unclear. Pandas cannot directly sense the nutritional quality of bamboo shoots but tend to sniff their food before consumption, inferring that odors inform their choice, which is then reinforced by the selection of positive and negative taste cues. To investigate the basis of selection, we observed the effects of 10 different bamboo species on feeding behavior, including food item selection, feeding frequency, portion size, food processing time per bite, and amount consumed per meal. Three of the bamboos were preferred, another four were consumed when the preferable bamboos were unavailable, and the remaining three were always rejected. We then characterized the volatile components of the bamboo leaves as well as the primary and secondary metabolites, allowing us, for the first time, to correlate feeding behavior with metabolomics. The three groups shared some volatile compounds but 21 volatiles were unique to the preferred leaves and appeared to confer sweet and fresh aromas, whereas the inedible leaves produced 20 unique volatiles that were pungent or floral, which appeared to discourage feeding. The three groups also shared many common nonvolatile metabolites, but pairwise comparisons revealed both qualitative and quantitative differences in metabolite abundance that resulted in the preferred leaves accumulating compounds associated with a sweet taste in humans (e.g., sugars), while the inedible leaves contained metabolites often associated with sour and bitter tastes (e.g., certain flavonoids and acids). Following attraction by certain volatiles, giant pandas may therefore consolidate their selection of leaves that are potentially more nutritious by consuming those with sweeter, less bitter and less sour tastes.

## 1. Introduction

Food selection by herbivorous mammals is an important component of foraging ecology but the underlying mechanisms are not clearly understood [1,2]. Proposed explanations include the nutrition hypothesis, plant secondary compound hypothesis, nutrient balance hypothesis, optimal foraging theory, and conditional odor avoidance hypothesis, which have been derived from direct field observations [2], clasp cage observations [3,4], buffet-style food choice experiments [5,6], the analysis of gastric bolus and stool [7], and predictive food selection models [8,9].

Giant pandas (*Ailuropoda melanoleuca*) provide a useful case study because they feed almost entirely on bamboo plants in the field, but select different combinations of species and organs according to the altitude and season in order to meet their nutritional needs [3,4,5,6,10]. Typically, giant pandas sniff bamboo shoots before feeding, suggesting that the odor of the bamboo can be used as a marker of nutritional quality. The olfactory system plays an important role in the detection of plant food sources because animals can smell the volatile compounds released by plants, and these act as cues to determine whether the plant is suitable for consumption. The reliance on odor rather than visual markers is supported by the ability of aging giant pandas to select appropriate food despite their de teriorating vision [3,8,9].

The gustatory system then provides additional feedback because metabolites that stimulate taste receptors often correlate with specific nutritional or anti-nutritional properties [11]. For example, beneficial or nutritional components are often associated with the perception of sweet (sugars), umami (glutamate) or salty tastes (minerals), whereas spoilage-related metabolites are often perceived as sour or bitter [12,13]. Many studies have shown that plant secondary metabolites can affect the foraging behavior of herbivores. For example, the ingestion frequency, ingestion rate and food intake of koala (*Phascolarctos cinereus*) decreased with increasing total levels of phloroglucinol in the leaves of *Eucalyptus robusta* [14], and the addition of eucalyptus phenols to leaves decreased food intake, food intake frequency, and food intake per meal in possum (*Trichosurus vulpecula*) [15]. Furthermore, red voles (*Microtus fortis*) preferentially chose tannin-free food at five levels of starvation [16] and similar results were reported for brown voles (*Lasiopodomys mandarinus*) and mice (*Mus musculus*) [17].

In order to determine whether the selection of bamboo by giant pandas is related to smell and taste perception, we presented four healthy giant pandas with bamboo leaves representing 10 species in buffet-style selection experiments and monitored their feeding behavior. We then used a combination of purge and trap-gas chromatography/mass spectrometry (PT-GC-MS) and high-performance liquid chromatography/mass spectrometry (HPLC-MS) to analyze the volatile and nonvolatile metabolomes of the bamboo species in order to determine which metabolites may be responsible for the observed behavioral responses. Our work provides insight into the sensory mechanisms underlying feeding behavior in giant pandas and will help to ensure that the nutritional needs of captive giant pandas are met to improve their quality of life and the performance of breeding programs.

## 2. Results

### 2.1. Analysis of Giant Panda Preferences for Bamboo

We observed distinct selective feeding patterns when monitoring the bamboo preference of giant pandas (Table 1). Based on the total food intake across four individuals, three bamboo species (*B*. *fargesii*, *P*. *japonica* and *I*. *tessellatus*) were classed as “preferred”, accounting for 81.2% of total consumption (9533 of 11,742 kg) (Appendix A). The consumption rates (food intake relative to feed volume) ranged from 37.5% to 41.0%, with a group average of 39.3%. Four *Phyllostachys* species (namely, *Ph. aureosulcata*, *Ph. vivax*, *Ph. propinqua* and *Ph. bissetii*) formed a secondary preference group, representing 18.7% of total intake (2199 kg), and were classed as “edible” because they were consumed primarily when preferred species were unavailable. Their average consumption rate was 8.7%. The last three species (*Ph. iridescens*, *Ph. praecox* and *Ph. parvifolia*) were classed as “inedible” because they were largely avoided, comprising only 0.085% of total intake (10 kg) with a near-zero consumption rate (0.03%).

A detailed temporal analysis of the feeding sequence was conducted to enrich the classification based on total consumption. The records consistently showed that giant pandas consumed preferred species first each day. The edible group species were ingested only after the preferred group was largely depleted, with minimal concurrent consumption. This sequential pattern, consistent across all four individuals, reinforces our hypothesis that the categories represent a distinct preference hierarchy driven by bamboo traits rather than random chance or availability. Quantitative analysis of feeding choices using *W_i_* confirmed significant selective feeding among the bamboo species (G-test, *p* < 0.01). The *W_i_* values revealed a clear preference hierarchy: the three preferred species (*B. fargesii*, *P. japonica* and *I. tessellatus*) had *W_i_* values significantly greater than 1 (confidence interval lower bound > 1), indicating strong preference. In contrast, the remaining seven species had *W_i_* values less than 1, indicating avoidance, with the lowest three species nearly untouched (*W_i_* ≈ 0). The feeding sequence was highly consistent across individual giant pandas, confirming that preferences were not random but probably driven by specific characteristics of the bamboo species. We therefore investigated the metabolites present in each species.

### 2.2. Analysis of Volatile Components

The analysis of leaves from 10 bamboo species by PT-GC-MS revealed 27~41 volatiles per species and 87 in total (Appendix A). This list comprised 19 alcohols, 18 alkenes, 16 alkanes, 7 heterocyclic compounds, 6 ketones, 6 aromatics, 4 ethers, 4 aldehydes, 3 acids, 3 esters, and 1 phenol. We identified 13 volatiles that were solely present in the three preferred bamboo leaves, nine that were unique to the edible leaves, and 15 that were unique to the inedible leaves. The most abundant volatiles in the preferred bamboo leaves were alcohols, alkenes and alkanes. In contrast, the most abundant volatiles in the edible and inedible leaves were alcohols, aromatics and alkenes (Appendix A).

Principal component analysis (PCA) was applied to all three groups of bamboo leaves. The volatiles from the preferred leaves were assigned to two principal components, the first representing 63% of the compounds. The volatiles from the edible leaves were assigned to three principal components, the first representing 49% of the compounds. Finally, the volatiles from the inedible leaves were assigned to two principal components, the first representing 59% of the compounds. The Venn diagram in Figure 1 shows that eight of the compounds were shared between all three groups, another 19 were shared between pairs of groups, and 48 were unique: 21 in the preferred leaves, 7 in the edible leaves and 20 in the inedible leaves. These compounds are listed in Table 2.

### 2.3. Identification of Primary and Secondary Nonvolatile Metabolites

To determine the components of the bamboo leaves that influence the taste preferences of the giant pandas, the primary and secondary metabolites present in the 10 bamboo species were extracted in methanol/water followed by identification and quantification by HPLC-MS. We prepared six replicates per species, making 60 samples in total. A data matrix containing the retention time, peak area, mass/charge ratio, and identification information was then prepared for post-processing and confidence analysis. The software extracted 16,208 peaks in positive ion mode and 18,516 in negative ion mode. Database screening identified 1238 of the positive ions and 1157 of the negative ions from various public metabolomic databases and KEGG (Table 3). We identified 22 different categories of metabolite in the KEGG database, the most diverse of which were amino acids, phospholipids and carboxylic acids (Figure 2a). These could be assigned to 15 major types of metabolic pathway, the most represented of which were amino acid metabolism, secondary metabolism, and carbohydrate metabolism (Figure 2b). Similarly, we identified 14 different categories of metabolites in the HMDB, the most abundant of which were lipids and lipid like molecules, phenylpropanoids and polyketide compounds, and organic oxygen compounds (Figure 3).

### 2.4. Correlation of Metabolic Profiles Across Bamboo Species

The abundance of metabolites in different samples was compared by correlation thermography to determine similarities within groups (preferred, edible and inedible) and differences between groups. The heat map of 60 bamboo samples shown in Figure 4 reveals, as expected, a strong degree of correlation between the six replicate samples of each species. It also shows a moderately high degree of correlation within groups, for example between samples HW and ZY, but in other cases a low correlation within groups, for example between samples HW and RC.

To resolve the structure of the data, we again used PCA to reduce the dimensionality, remove noise and redundancy, and transform multiple indicators into a few independent comprehensive indicators that contain most of the original information. We identified two principal components, the first explaining 23.9% of the variation and the second explaining 14%. This separated the species into two main clusters, the first representing the preferred bamboo leaves and the second representing both the edible and inedible leaves, with overlapping confidence ellipses indicating the distribution of the real samples with 95% confidence (Figure 5a). We applied linear discriminant analysis (LDA) to optimize the responses, revealing a first linear discriminant explaining 19.3% of the variation and a second explaining 12.5%, reflecting a cumulative contribution of 31.8%, which better presented the overall information from the sample (Figure 5b). Again, the preferred bamboo leaves were well separated from the rest of the samples, but LDA better separated the edible from the inedible samples, with only a small overlap (manifested in LD2). Even so, these results indicate that the edible and inedible samples have some metabolic similarities.

By comparing the metabolites in each bamboo species, we found that 1210 compounds were shared among all three groups. However, we also identified seven unique cations and 14 unique anions in the preferred bamboo leaves, three unique cations and four unique anions in the edible leaves, and eight unique cations and five unique anions in the inedible leaves (Table 4). The relationships are shown as Venn diagrams in Figure 6. The “unique” metabolites identified here are defined by their detection above a defined threshold within one group and their absence below that threshold in others, as described in the Methods.

### 2.5. Quantitative Analysis of Metabolites Differing in Abundance Between Bamboo Groups

In addition to the identification of metabolites that are present in one group of bamboo species and absent in others, we also investigated quantitative differences in the abundance of cations and anions between the groups, using a difference threshold of three-fold at a significance of *p* < 0.05. These differences are shown as volcano plots in Figure 7 and the corresponding lists of quantitative values are presented in Appendix A. We detected 1214 cations (152 identified) and 1644 anions (168 identified). Pairwise significant differences in abundance are shown in Table 5 and are summarized as a Venn diagram in Figure 8. Heat maps showing the clustering of metabolites with significant pairwise differences in abundance reveal a clear oppositional trend between the preferential bamboo samples and the other two groups of samples, with less profound differences between the edible and inedible bamboo shoots (Appendix A).

We investigated the KEGG pathways associated with the metabolites that differ in abundance in pairwise comparisons between bamboo groups, and found that most of the metabolites were associated with the primary categories “metabolism” and “environmental information processing”. Breaking these down into secondary classifications revealed similarities between the pairwise comparisons but also some differences (Figure 9). For example, although the top classification in all three comparisons was “secondary metabolism”, the relative importance of carbohydrate and amino acid metabolism was much higher in the preferred vs. edible comparison than the other two pairings. There were also fewer pathways associated with the edible vs. inedible comparison than the other two pairings. Overall, 30 of the 240 differential metabolites in the preferred vs. inedible comparison were allocated to nine KEGG pathways (Figure 9a), 35 of 182 differential metabolites in the preferred vs. edible comparison were allocated to nine KEGG pathways (Figure 9b), and 11 of the 68 differential metabolites in the edible vs. inedible comparison were mainly allocated to six KEGG pathways (Figure 9c).

The enrichment of a KEGG pathway is the ratio of the number of metabolites enriched in the pathway to the number of metabolites annotated in the pathway. The main pathways enriched in the preferred vs. inedible bamboo samples were the biosynthesis of flavonoids, flavones and flavonols, phenylalanine, tyrosine and tryptophan, as well as the metabolism of amino sugars, nucleotide sugars, ascorbate and alderate. The enrichment of flavonoid biosynthesis showed the greatest significance (Figure 10a). The same biosynthesis pathways were enriched in the preferred vs. edible bamboo samples, with flavonoid biosynthesis again showing the greatest significance, but the metabolism of amino sugars and nucleotide sugars was displaced by starch and sucrose metabolism (Figure 10b). Interestingly, when comparing the edible vs. inedible bamboo samples, the only enriched pathways were flavone and flavonol biosynthesis as well as proline and arginine metabolism (Figure 10c).

Finally, we mapped our quantitative metabolic dataset onto HMDB v4.0, allowing the successful annotation of 197/240 differential metabolites in the preferred vs. inedible comparison, 150/182 in the preferred vs. edible comparison, and 58/68 in the edible vs. inedible comparison (Figure 11). The distribution of metabolites at the superclass, class and subclass levels was similar in all three pairwise comparisons, dominated by lipids and lipid-like molecules (~34–40%), phenylpropanoids and polyketides (~22–26%), organic oxygen compounds (~8–13%) and organohetrocyclic compounds (~7–10%).

## 3. Discussion

Giant pandas eat bamboo leaves, poles, shoots and other parts, and prefer tall and thick new shoots [8]. However, different local populations will choose distinct bamboo species at different altitudes and will consume different bamboo parts depending on the season, such as young shoots and leaves in summer and autumn, but branches, leaves and poles in winter [6]. This reflects the differing protein, fat, sugar and fiber content of different bamboo parts [18]. Sufficient nutrition therefore requires giant pandas to choose bamboo species and parts according to need, but the mechanism of selection is not yet understood.

The analysis of seasonal changes in feeding habits has shown that giant pandas in the Qinling Mountains feed on *B. fargesii* shoots in April, because they are rich in nitrogen and phosphorus. But as the shoots age and harden, and their nutritional value decreases, the animals migrate to higher altitudes to find *Fargesia qinlingensis* T. P. Yi & J. X. Shao shoots, which remain rich in nitrogen and phosphorus later in the year. In July, they begin to consume *F. qinlingensis* leaves, which are rich in calcium. In August, females return to lower altitudes to give birth, and feed on *Bashania bambusa* leaves in February. From March, the giant pandas seek out new poles of *B. fargesii* and consume them at the same time as the leaves [8]. This pattern is repeated annually to maintain basic nutritional requirements.

The feeding preference of giant pandas is usually measured by calculating the Vanderploeg and Scavia selection coefficient (*W_i_*) and selection index (*E_i_*) for different bamboos [19,20]. Giant pandas in the Qinling Mountains prefer *F. qinling* and *Dendrocalamus bashania*, whereas those in the Minshan Mountains select *Fargesia denudata* T. P. Yi, *Fargesia obliqua* T. P. Yi and *Fargesia nitida* (Mitford ex Stapf) P. C. Keng ex T. P. Yi. Giant pandas in the Qionglai Mountains prefer *Fargesia frigidis* T. P. Yi, those in the Xiangling Mountains prefer *Yushania cava* T. P. Yi and those in the Liangshan Mountains preferentially select *Yushania glauca* T. P. Yi & T. L. Long, *Yushania mabianensis* T. P. Yi, *Chimonobambusa szechuanensis* (Rendle) P. C. Keng, *Qiongzhuea tumidinoda* J. R. Xue & T. P. Yi and *Qiongzhuea macrophylla* J. R. Xue & T. P. Yi [19]. The preferred species are therefore dependent on multiple factors, such as the region, altitude and climate. However, there may be some common major components among these species that underlie the observed preferences, and the analysis of bamboo metabolites could provide insight into the selective process. We therefore designed a buffet-style feeding experiment in which four giant pandas were presented with 10 different species of bamboo on consecutive days, and we weighed the different foods before and after each meal to see how much of each sample was consumed. The 10 species were chosen based on their abundance (no rare species) and suitability for large-scale cultivation in Beijing to meet the giant panda’s nutritional demands. We then analyzed the volatile and non-volatile metabolites in each species in order to find correlates with food selection.

The olfactory system plays an important role in the detection of plant food sources because animals can smell the volatile compounds released by plants, and these often act as cues to determine whether the plant is suitable for consumption. We used PT-GC-MS to detect and identify the volatiles released by bamboo leaves, and found that the eight volatile compounds we detected were associated with five kinds of aroma: green, leaf, fruit, musk and wine [21,22]. Although PCA and LDA indicated considerable metabolic overlap between the edible and inedible groups (Figure 5), their consumption rates differed vastly. This suggests that the critical factor may not be the overall metabolic profile but rather quantitative differences in specific deterrent compounds. Key bitter-associated metabolites (e.g., specific flavonoids) might exceed a behavioral rejection threshold in the inedible group. Furthermore, initial olfactory rejection, driven by the unique pungent or floral volatiles found in the inedible leaves (Table 3), may prevent pandas from sampling these species, thereby consolidating the difference in consumption. We identified 21 volatile components associated solely with the preferred bamboo leaves (Table 2), six of which are associated with green and leafy aromas (3-methyl-2-butanol, (1S)-2-methyl-1-butanol, (R)-(–)-2-pentanol, 2,2,4,4-tetramethyl-1,3-cyclobutanediol, ethanol, and 1,3-propanediol), three with a fresh aroma (camphene, 1,7,7-trimethyl-tycloheptane, and (1S)-1,7,7-trimethyl-dicycloheptan-2-one), as well as sec-butyl ester-cyanic acid (fruity) and 2-ethyl-furan (sweet and coffee-like). Other volatile components were associated with pungent, unpleasant odors such as (1R)-2,6,6- trimethylbicycloheptan-2-ene, α-p-ene, 2-methyl-l-butene, *N*,*N*,*N*′,*N*′-tetramethyl- methylenediamine (ammonia-like), (1S)-6,6-dimethyl-2-methylene-dicycloheptane, 1-benzyl-3-amino-4-cyano-3-pyrroline, carbonyl sulfur (rotten eggs), hexamethyldisiloxane, and *N*-dimethylaminomethyl tert-butyl isopropyl phosphine. The seven volatiles unique to the edible bamboo leaves were 1-vinyl-4-methoxy-benzene (slightly sweet anise aroma) and six rather pungent compounds (2,4-dimethyl-1,3-pentadiene, (*Z*)-1,3-pentadiene, 2,2-dimethyldecane, 2-butenyl-hydrazine, 2-methyl-3-butan-2-ol and the petroleum-like aroma of propylcyclopropane). The 20 volatiles unique to the inedible bamboo leaves included the green and leafy aromas of 2-pentyl-furan and dimethylsulfide, the wine-like aroma of 1-undecylene, the floral aromas of 3,4-dimethyl-2-hexanone and 3-isopropenyl-5,5-dimethyl-cyclopentene, and the aromatic ether odors of 1-[(2-methyl-2-propyl)oxy]-butane and 1-methylbutyl-ethylene oxide. The other components were associated with pungent odors, namely cyclohexane, dichloromethane, 2-methoxy-ethanol, 2-methyl-1,3-butadiene, 3-ethyl-2,2-dimethyl-pentane, (*Z*)-2-pentene, formic acid, 1,3-dioxazole-2-one, *N*-propyl-3,4-methylenedioxyamphetamine (amine odor) tri(trimethylsilyl)borate, fluoropropene, 2-fluoropropene and octamethyl-cyclotetrasiloxane. The three groups of bamboo therefore shared green, leafy, fruity, musky and wine-like odors but the preferred species were characterized by additional sweet and fresh aromas (less prevalent but still present in the edible species) whereas aromatic floral odors were more prevalent in the inedible species and may provide cues for the giant pandas to avoid. One limitation of our analysis of volatiles is that the sample preparation method (pulverization and heating) might generate a volatile profile differing from that emitted by an undisturbed leaf. We attempted analysis at lower temperatures (35 and 45 °C) but this did not generate sufficient volatiles for accurate detection. High temperatures enhance the release of volatiles and should at least approximate the profiles detected by giant pandas presented with fresh vegetation. However, future studies based on dynamic headspace sampling of intact leaves could more closely approximate the olfactory cues available to foraging pandas.

In mammals, many essential nutrients influence taste perception, including sugars, amino acids and nucleotides [23]. Taste receptors are G-protein coupled receptors that can be divided into two major families, responsible for the perception of sweetness (TAS1R) and bitterness (TAS2R), respectively [24,25]. There are three sweetness receptors (TAS1R1, TAS1R2 and TAS1R3) which function predominantly as heterodimers: TAS1R2/TAS1R3 detects sweet tastes by binding sugars and sweeteners [24,26], whereas TAS1R1/TAS1R3 detects umami mainly by binding to glutamate [24,26]. In contrast, mammals possess a much larger number of bitterness receptors, reflecting the diverse ligands that are perceived as bitter, including alkaloids, flavonoid glycosides and inorganic salts [27,28]. The number ranges from fewer than 10 to more than 60, and is generally higher in herbivores/omnivores than carnivores because plant tissues contain a larger number of bitter-tasting toxic compounds than animal tissues, suggesting there has been selective pressure for the diversification of the TAS2R family to enable herbivores to avoid toxic plants [29,30,31].

Giant pandas represent an interesting case study because they are herbivores that diverged from a carnivorous ancestor only 43 Mya, and therefore belong to the order Carnivora [32]. More than 90% of the food consumed by giant pandas is bamboo, which is known to contain abundant bitter secondary metabolites such as phenolics, terpenoids, alkaloids, flavonoids and cyanogenic glycosides [33,34], many of which we detected in our metabolomic analysis of 10 bamboo species. It therefore seems likely that giant panda taste buds will have evolved to sense bamboo tissues that are suitable as food while rejecting food sources that cause harm. Interestingly, a comprehensive analysis of *TAS2R* genes in nine Carnivora has shown that the giant panda (as well as the closely related red panda, *Ailurus fulgens*) possesses 16 intact *TAS2R* genes, which is more than truly carnivorous relatives in the same order, such as polar bears, wolves, tigers and cheetahs, with 10–14 [35]. This is still far fewer than some other herbivores and omnivores, such as humans with 43 intact *TAS2R* genes [11], but giant pandas have accumulated many useful mutations over a relatively short evolutionary timespan, helping them detect metabolites in bamboo and other foods. The expansion of the *TAS2R* gene family in pandas may therefore have refined the sense of taste to facilitate this process.

To identify metabolites in bamboo leaves that might explain the preference for different species, we fractionated methanol/water extracts by HPLC-MS and identified peaks that were unique to the preferred, edible and inedible leaves, as well as ions that showed a significant quantitative difference in abundance. These were mapped onto KEGG pathways, allowing us to annotate the metabolites identified in each group. We found that the biosynthesis of secondary metabolites was the most enriched pathway in all three pairwise comparisons, but that carbohydrate and amino acid metabolism were also strongly represented. Carbohydrates and amino acids are primary metabolites and important dietary components that indicate the nutritional properties of the bamboo, so it was interesting to note the correlation with flavonoid biosynthesis in the inedible bamboo leaves, which may act as a bitter-tasting correlate of anti-nutritional properties. Heat maps of each pairwise comparison revealed similar trends in differential metabolite abundance in the preferred and edible bamboo leaves, with the opposite trend observed in the inedible leaves. The increase in the content of flavonoids is likely to increase the bitter taste of the bamboo leaves, together with the amino acids phenylalanine, tyrosine and tryptophan. The increase in starch metabolism and the accumulation of sucrose, amino sugars and nucleotide sugars are likely to make the leaves sweeter, whereas ascorbate and alderate metabolism may increase their sourness. Having been attracted by certain volatiles, giant pandas may therefore consolidate their selection of nutritionally beneficial leaves by consuming leaves with a metabolic profile suggestive of a sweeter and less bitter/sour taste, although we acknowledge that our analysis is based on human taste associations. The sensory descriptors of volatiles (e.g., sweet, pungent and floral) are assigned based on human sensory databases and the corresponding literature [21,22] and are used as a reference framework. The actual perception of these complex odor blends by giant pandas may differ, but may nevertheless provide rules facilitate the selection of preferred bamboo species for captive giant pandas. This study focused on volatile and polar/semi-polar metabolites, excluding important biochemical classes such as lipids and complex structural carbohydrates (e.g., fibers). These components play key roles in nutrition and palatability (e.g., via texture and energy content) and their interaction with the identified chemical cues is an important avenue for future research. It is important to note that this study did not include proximate nutritional analysis (e.g., crude protein or fiber content). Thus, the link between the identified chemical profiles and actual nutritional value is inferred based on the known associations of certain metabolite classes (e.g., sugars an amino acids) with nutritional quality [18]. Our conclusions therefore identify these sensory cues as putative, rather than confirmed, markers of nutritional status. Our findings are derived from a controlled study with a limited number of captive pandas. The bamboo species offered were selected based on availability and cultivability in Beijing, and the preferences we observed therefore cannot be extrapolated directly to the complex foraging ecology of wild panda populations across different mountain ranges. Instead, this study should be viewed as providing a foundational hypothesis that specific volatile and non-volatile metabolites are key cues, which can be tested in future field studies with wild pandas.

## 4. Materials and Methods

### 4.1. Bamboo Sampling

Samples of 10 bamboo species were sourced from Beijing Botanical Garden: *Bashania fargesii* (E. G. Camus) P. C. Keng & T. P. Yi, *Pseudosasa japonica* (Siebold & Zucc. ex Steud.) Makino ex Nakai, *Indocalamus tessellatus* (Munro) P. C. Keng, *Phyllostachys aureosulcata* McClure, *Phyllostachys vivax* McClure, *Phyllostachys propinqua* McClure, *Phyllostachys bissetii* McClure, *Phyllostachys iridescens* C. Y. Yao & S. Y. Chen, *Phyllostachys praecox* C. D. Chu & C. S. Chao and *Phyllostachys parvifolia* C. D. Chu & H. Y. Chou. Samples were collected on 20 August, 4 September and 10 September 2020. The samples were divided into three parts, for feeding, PT-GC-MS and HPLC-MS.

### 4.2. Selection of Experimental Animals

Four captive giant pandas in vibrant health from Beijing Zoo took part in a buffet feast experiment. Our subjects included two 9-year-old males (A and B), one 23-year-old male (C), and one 16-year-old female (D).

### 4.3. Buffet-Style Feeding Experiment

The giant pandas were fed according to their accustomed schedule at 1:30 p.m. every day for 4 days with ~1–3 kg of each type of bamboo leaves. The leaves were weighed before and after each meal to determine how much was consumed. We then calculated the forage ratio selection index *W_i_* (Equation (1)) to confirm the principle of discrimination [36].*W_i_* = *O_i_*/*P_i_*(1)
where *W_i_* is the tendency to feed on the first bamboo species, *O_i_* is the proportion of the first bamboo species in the diet, and *P_i_* is the proportion of the first bamboo species in the total amount of bamboo. A G-test was then used to determine the significance of bamboo species selection [37] based on the null hypothesis that selection was random (Equation (2)).(2)$$\chi^{2}=2\sum_{i=1}^{n}\left[u_{i}ln\left(\frac{u_{i}}{Up_{i}}\right)+m_{i}ln\left(\frac{m_{i}}{\left(m_{i}+u_{i}\right)M/(U+M)}\right)\right]$$
where *u_i_* is the weight of the first bamboo selected, *m_i_* is the weight of the first bamboo consumed, χ^2^ is the chi-square value with (*n* − 1) degrees of freedom, and *n* is the number of resource types. When the degree of freedom (*df*) was *n* − 1 at a probability of *p* ≤ 0.05, the giant panda has selectivity for this type of bamboo. To confirm whether giant pandas prefer or avoid specific bamboo species, we calculated the confidence interval [38] using *W_i_* ± *Z_a_Sw_i_* (Equation (3)).*Sw_i_* = [(1 − *O_i_*)/*UO_i_* + (1 − *P_i_*)/*MP_i_*]^1/2^(3)
where *Sw_i_* is the standard deviation of the bamboo utilization rate. The normal distribution of *Z_a_* was adjusted by Bonferroni correction and the significance level α was reduced to α/2*n* to obtain the corresponding value of *Z_a_*. A lower bound of the *W_i_* confidence interval >1 indicated preference for a bamboo species (high feeding probability, large feeding amount or proportion) whereas an upper bound < 1 indicated avoidance (low feeding probability, small feeding amount or proportion. A value of 1 represents random selection. The G-test for significance of selection was applied, and the resulting p values were considered significant at *p* ≤ 0.05. The confidence intervals for *W_i_* were adjusted for multiple comparisons using the Bonferroni method.

At the end of each experiment, the average feeding amount for each bamboo species was calculated in groups without considering differences between the giant pandas, and the G-test for significance was repeated. As above, *df* = *n* − 1 and *p* ≤ 0.05 was considered to indicate selective feeding, and the tendency to favor or disfavor particular species was determined by calculating the confidence interval of *W_i_*.

### 4.4. Determination of Volatile Components

The volatile components of bamboo leaves were characterized by PT-GS-MS as previously described [39]. Briefly, leaf samples 50 ± 10 cm in length were cut from 1 to 2-yearfrom 1–2-year-old bamboo shoots and stored at −20 °C. They were washed, air dried and pulverized in a DFT-50 Chinese medicine mill. We then mixed 2 g of powder with 8 mL ultrapure water and heated to 80 °C in a water bath prior to analysis. The samples were heated to 180 °C to release volatile components into a collector tube containing a mixed adsorbent of equal parts Tenax, silica gel and activated carbon. Desorption was achieved by heating to 190 °C for 30 s while purging with nitrogen gas using an Eclipse 4660 automatic injector connected to a QP2010 GC/MS device (Shimadzu Company, Kyoto, Japan). The GC instrument was fitted with an RTX502.2 capillary column (60 m × 0.32 mm × 1.8 μm). The vaporization chamber temperature was 220 °C, and the programmed temperature started at 30 °C (hold for 5 min) then increased at 10 °C/min to 190 °C and at 20 °C/min to 250 °C with a hold at 3 min. The MS instrument was fitted with an electron bombardment ionization source at 70 eV. The ion source temperature was 200 °C and the interface temperature was 220 °C. The electron multiplier voltage was 1.02 kV. Data were acquired in full scan or selected ion scan mode (SIM) over the range 45–350 AMUs. This PT-GC-MS method is a standardized approach for profiling plant volatiles, but it should be noted that it may not perfectly replicate the odor profile sensed by an animal from an intact, fresh leaf in a natural foraging context.

### 4.5. Analysis of Bamboo Metabolites

Bamboo leaves were ground to a fine powder under liquid nitrogen and stored at −80 °C. We then mixed 50 mg of each sample with 400 μL methanol/water (4:1 *v*/*v*) containing 0.02 mg/mL internal standard (l-2-chlorophenylalanine) in a 2 mL centrifuge tube and prepared extracts using a grinding bead (6 min, −10 °C, 50 Hz) followed by sonication (30 min, 5 °C, 40 kHz). After centrifugation (13,000× *g*, 15 min, 4 °C), the supernatant was transferred to an Acquity UPLC device fitted with an HSS T3 column (100 mm × 2.1 mm, 1.8 μm; Waters, Milford, Massachusetts, USA) and fractionated in a gradient of mobile phase A (0.1% formic acid in 95:5 water/acetonitrile) and mobile phase B (0.1% formic acid in 47.5:47.5:5 acetonitrile/isopropanol/water) at 40 °C. Fractions were automatically injected into a Q Exactive HF-X mass spectrometer (Thermo Fisher Scientific, Waltham, MA, USA) fitted with an ESI source and mass spectra were collected in positive and negative ionization modes. The raw data were imported into Progenesis QI (Waters) for baseline filtering, peak identification, integration, retention time correction, and peak alignment. Quality control (QC) was performed using pooled samples from all bamboo extracts. These QC samples were injected at regular intervals throughout the analytical sequence to monitor instrument stability. Data processing included baseline filtering, peak identification, integration, retention time correction, peak alignment, and normalization. Metabolite identification confidence was ranked according to the COSMOS standards: Level 1 (confirmed by standard) and Level 2 (putatively annotated based on spectral similarity to public libraries). A data matrix containing retention time, peak area, mass/charge ratio, and identification information was prepared for post-processing and confidence analysis. The peak characteristics and spectral data were then used for metabolite annotation by screening public databases, including the Human Metabolome Database (HMDB) (http://www.hmdb.ca/, accessed on 8 Sptember 2021), MELTIN G2 (https://metlin.scripps.edu/, accessed on 10 Sptember 2021), LIPID MAPS (https://www.lipidmaps.org/, accessed on 9 Sptember 2021), and the Kyoto Encyclopedia of Genes and Genomes (KEGG) as well as plant-specific databases such as KNApSAcK, the Plant Metabolic Network (PMN, https://www.plantcyc.org, accessed on 12 Sptember 2021), and LIPID MAPS.

For the analysis of “unique” metabolites in Venn diagrams, a compound was considered present in a bamboo group if it was detected in at least five of the six biological replicates for a species within that group, with a peak area greater than three times the background noise level. Absence indicates that the compound was not reliably detected above this threshold and may not imply absolute absence.

## Figures and Tables

**Figure 1 plants-14-03856-f001:**
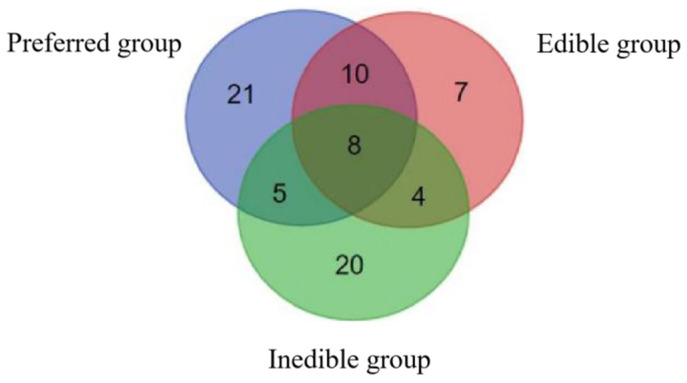
Venn diagram showing compounds unique to the preferred, edible and inedible bamboo leaves, and the compounds shared between those groups.

**Figure 2 plants-14-03856-f002:**
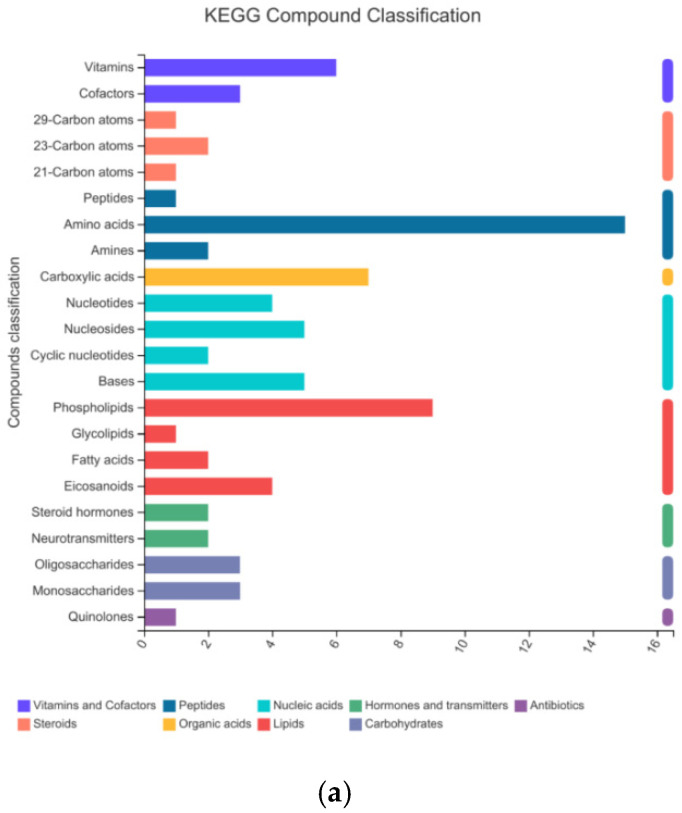
Metabolites from bamboo leaves identified in the KEGG database assigned by (**a**) compound classification and (**b**) metabolic pathway classification.

**Figure 3 plants-14-03856-f003:**
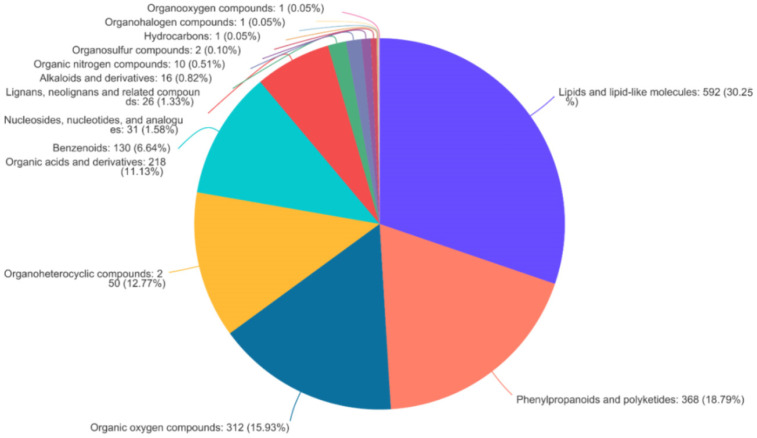
Metabolites from bamboo leaves annotated in HMDB v4.0 assigned by compound classification.

**Figure 4 plants-14-03856-f004:**
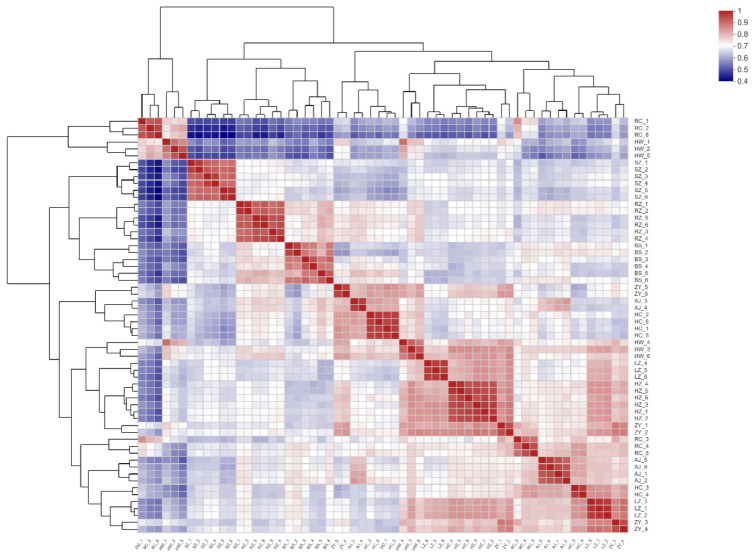
Heat map showing metabolic correlation between bamboo species. BS1-6, RZ1-6 and SZ1-6 are the preferred leaves, HC1-6, HW1-6, ZY1-6 and RC1-6 are the edible leaves, and HZ1-6, LZ1-6 and AJ1-6 are the inedible leaves. The more red the coloring, the greater is the correlation.

**Figure 5 plants-14-03856-f005:**
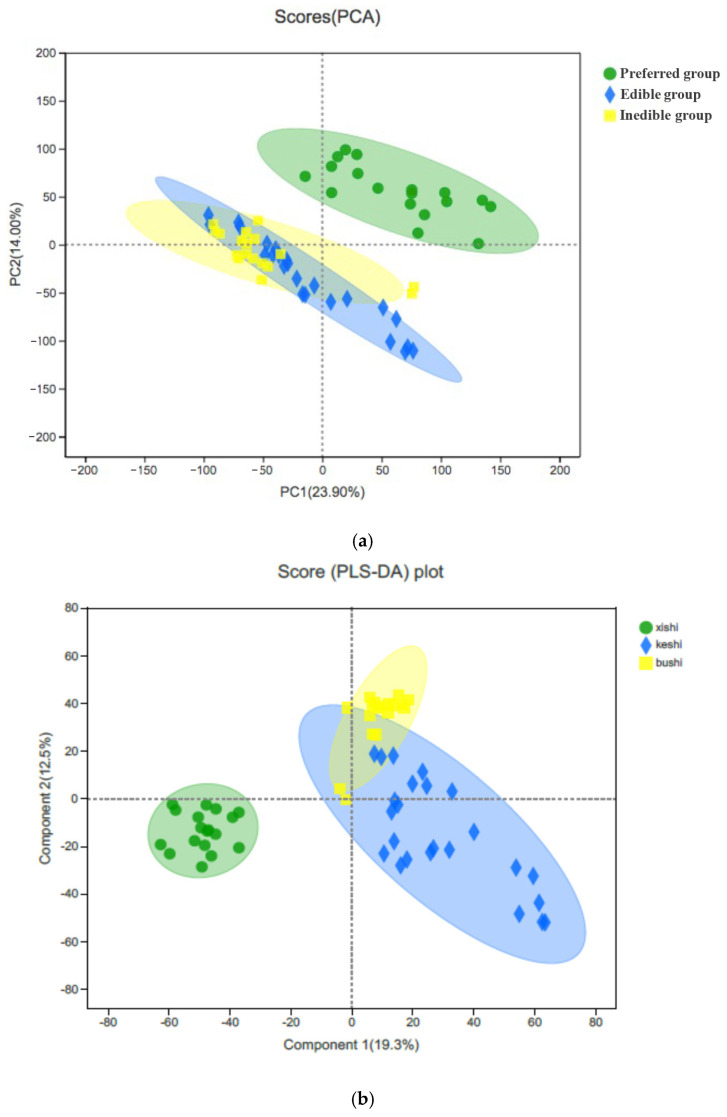
Clustering of the metabolites extracted from 10 bamboo species by (**a**) principal component analysis and (**b**) linear discriminant analysis. The plots show the separation of the preferred leaves (green) from a combined cluster encompassing both the edible leaves (blue) and inedible leaves (yellow), the latter being better separated by LDA than PCA.

**Figure 6 plants-14-03856-f006:**
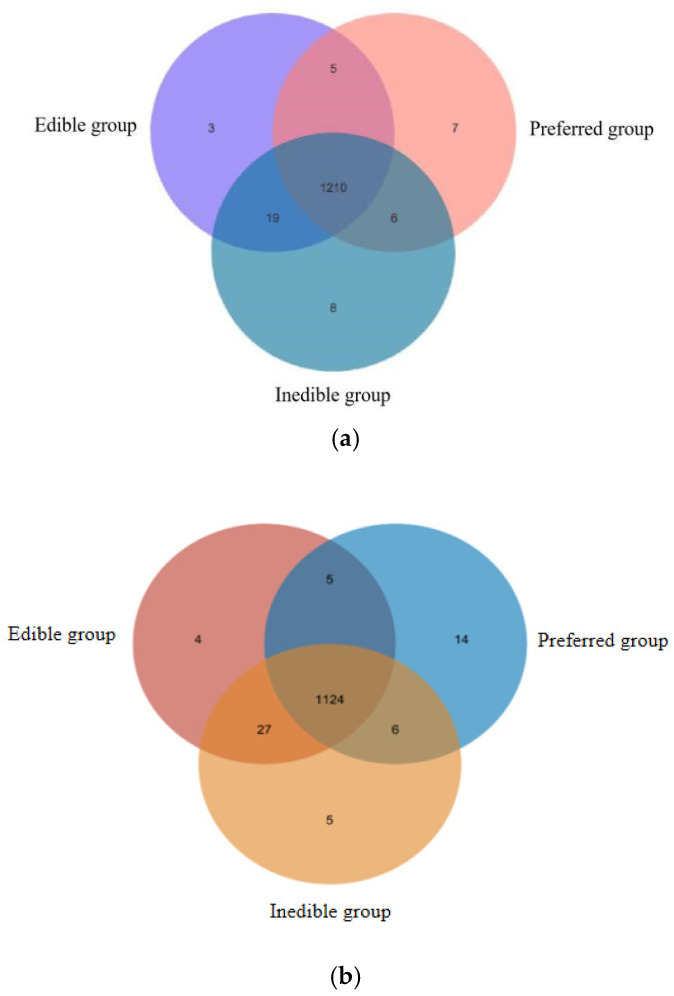
Distribution of metabolites detected in the preferred, edible and inedible bamboo leaves represented by (**a**) cations and (**b**) anions detected by mass spectrometry.

**Figure 7 plants-14-03856-f007:**
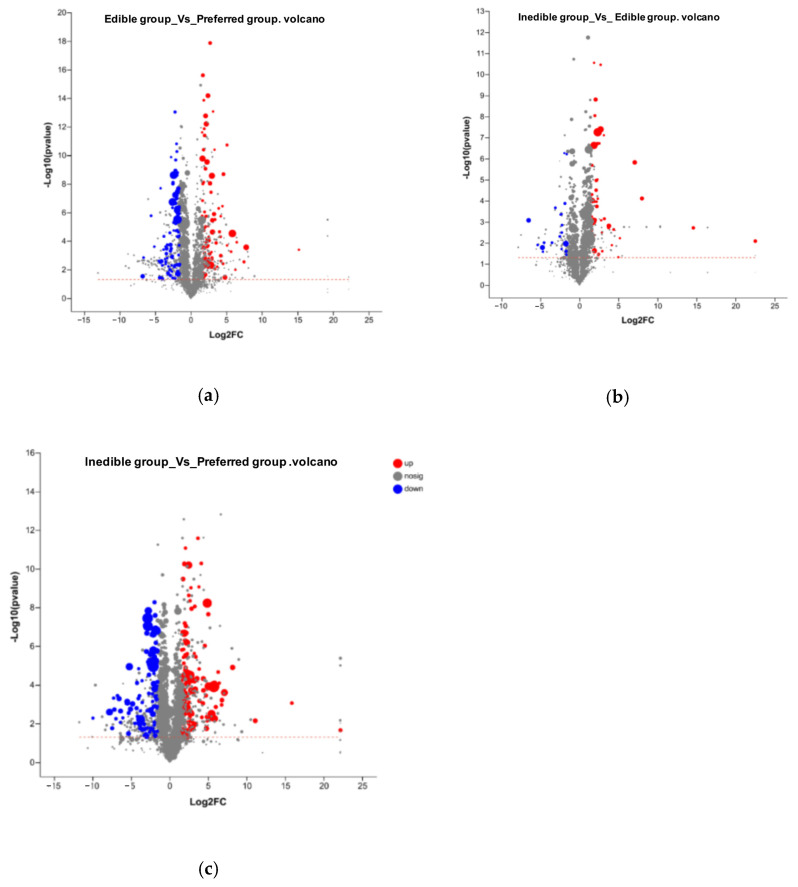
Volcano plots showing an overview of the quantitative differences in metabolite abundance between (**a**) preferred vs. edible groups, (**b**) edible vs. inedible groups, and (**c**) preferred vs. inedible groups. The *x*-axis shows the log_2_ fold-change between groups and the *y*-axis shows the statistical test of the fold-change (log_10_ *p* value) with higher vales indicating greater significance. Each point in the figure represents a specific metabolite. The size of each point represents the Vip value and the color indicates relative accumulation (red) or depletion (blue). See Appendix A for the corresponding datasets.

**Figure 8 plants-14-03856-f008:**
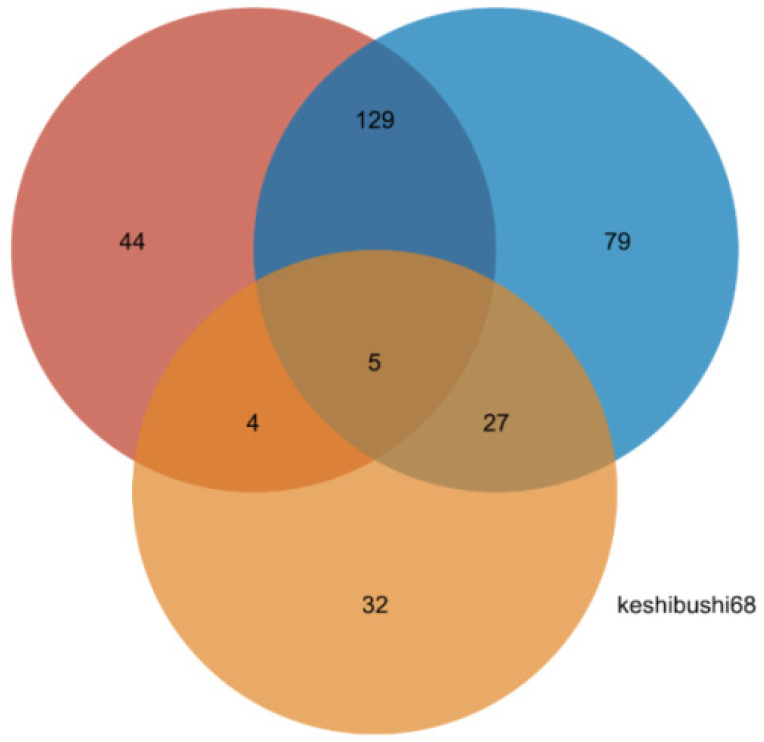
Distribution of metabolites that differ quantitatively in abundance in pairwise comparisons between the preferred, edible and inedible bamboo leaves (numbers represent totals of identified cations and anions.

**Figure 9 plants-14-03856-f009:**
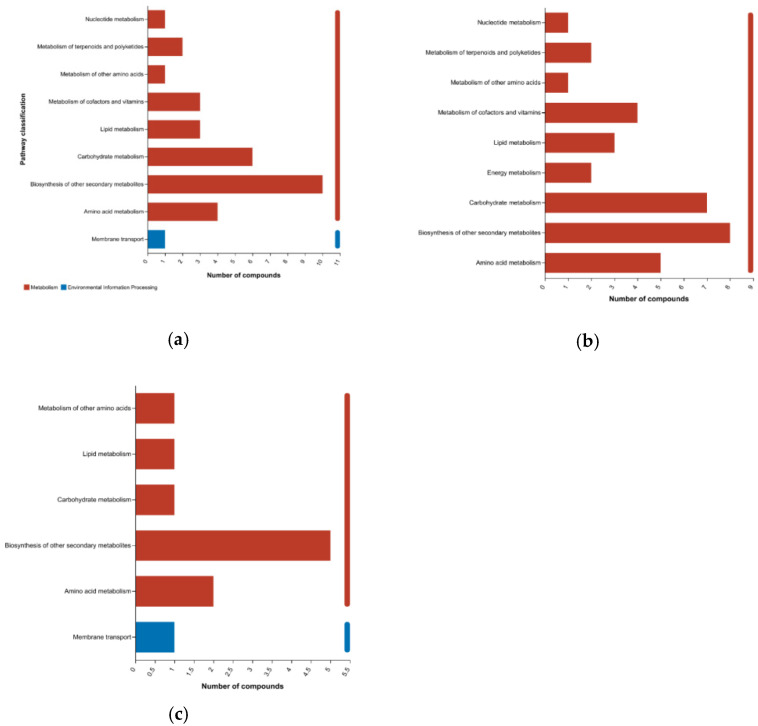
Allocation of metabolites differing in abundance between bamboo groups to KEGG pathways. The panels show differences in metabolite allocation between (**a**) preferred vs. inedible, (**b**) preferred vs. edible, and (**c**) edible vs. inedible comparisons. The *y*-axis shows the secondary classification of KEGG metabolic/environmental pathways and the *x*-axis shows the number of metabolites allocated to each pathway.

**Figure 10 plants-14-03856-f010:**
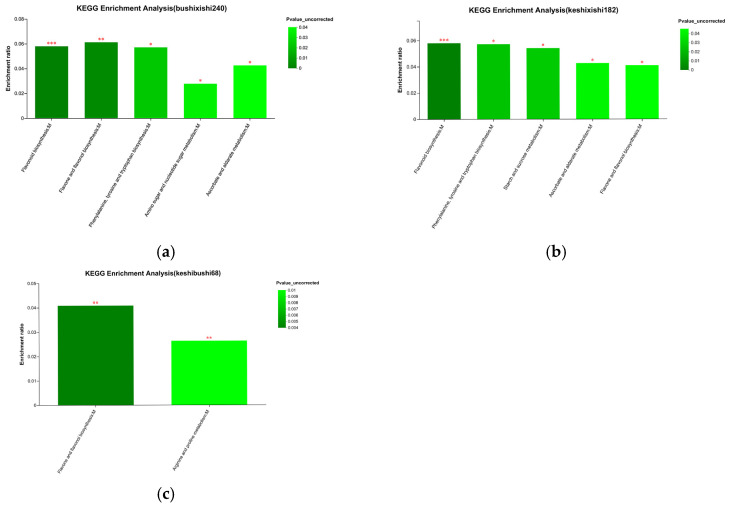
KEGG pathway enrichment of metabolites differing in abundance in pairwise comparisons between bamboo groups. Data are shown for (**a**) preferred vs. inedible, (**b**) preferred vs. edible and (**c**) edible vs. inedible comparisons. The color gradient indicates the significance of enrichment (darker = more significant). Significance is indicated as follows: *** *p* (FDR) < 0.001, ** *p* (FDR) < 0.01, * *p* (FDR) < 0.05.

**Figure 11 plants-14-03856-f011:**
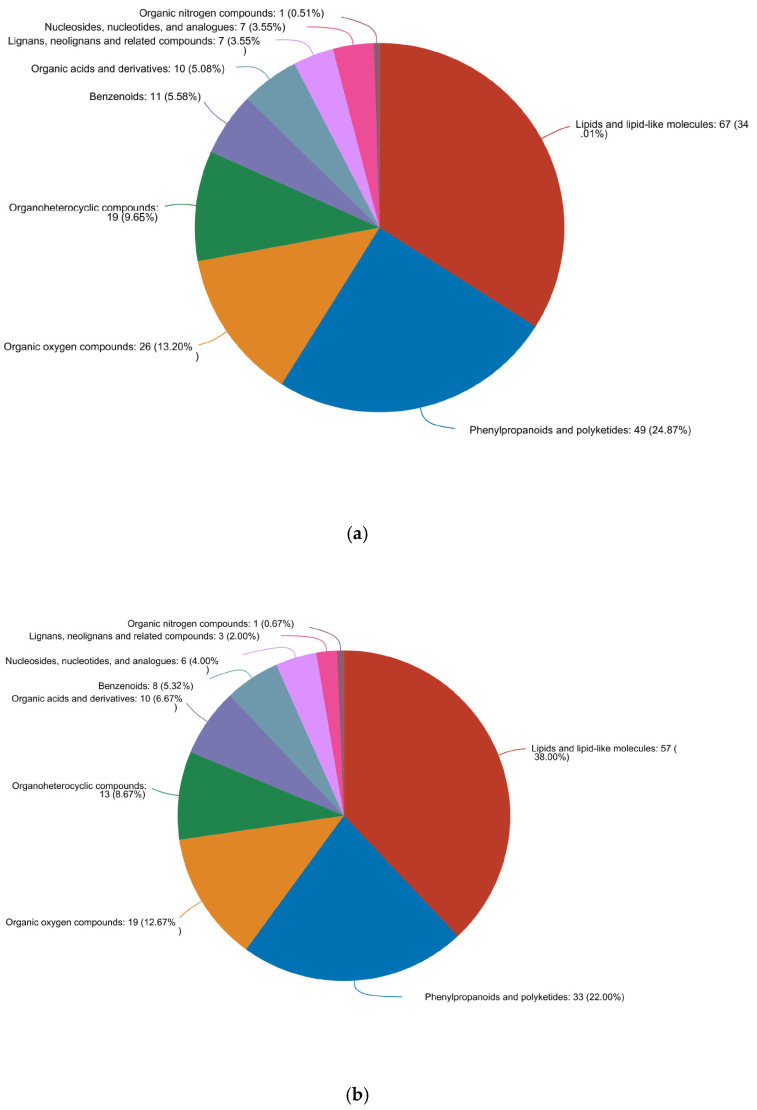
Annotation and classification of different metabolites in HMDB v4.0. Data are shown for (**a**) preferred vs. inedible, (**b**) preferred vs. edible and (**c**) edible vs. inedible comparisons. More than 75% of the metabolites in all three comparisons fall into four major superclass/class/subclass groups.

**Table 1 plants-14-03856-t001:** Analysis of giant panda preferences for bamboo species.

Species	A			B			C			D		
	Feed Volume	Food Intake	Preference Tendency	Feed Volume	Food Intake	Preference Tendency	Feed Volume	Food Intake	Preference Tendency	Feed Volume	Food Intake	Preference Tendency
	(kg)	(kg)		(kg)	(kg)		(kg)	(kg)		(kg)	(kg)	
*B. fargesii*	1740	445	P	1850	675	P	1925	680	P	2088	1316	P
*I. tessellatus*	1280	95	P	1663	650	P	1742	472	P	1734	1193	P
*P. japonica*	2180	620	P	2848	1918	P	2510	1125	P	2661	344	P
*Ph. bissetii*	1314	23	A	1078	158	A	1596	146	A	1708	125	A
*Ph. aureosulcata*	1164	45	A	700	135	A	1197	151	A	1684	133	A
*Ph. vivax*	1920	68	A	2476	421	A	1817	153	A	1963	90	A
*Ph. propinqua*	1503	148	A	1971	164	A	1737	79	A	1515	160	A
*Ph. parvifolia*	3055	5	R	2208	0	R	2635	5	R	2730	0	R
*Ph. iridescens*	2250	0	R	2253	0	R	2040	0	R	2480	0	R
*Ph. praecox*	2900	0	R	2803	0	R	2535	0	R	2555	0	R

Note: P indicate preference, A indicate available, R indicate refuse.

**Table 2 plants-14-03856-t002:** Volatiles found uniquely in the preferred, edible and inedible bamboo leaves, and the compounds shared between those groups.

Distribution	Number	Volatile Components
Common to all three groups	8	1,3-pentadiene, 2-pentanol, 2-methylbutan-1-ol, 1-hexanol, sec-hydroxymethyl thiobenzoate, trimethylsilanol, 1,4-dioxene, 3-methyl-1-butanol
Preferred + edible	10	1,2:5,6-dihydrogalactol, hexamethylcyclotrisiloxane, 2-penten-1-ol, 1-ethyl-4-methoxybenzene, methoxymethyloxirane, 5-octyl-3-ol, acetone, 2-hexene, octadecyltrisiloxane, 2-methyl-2-butene
Preferred + inedible	5	2-methylfuran, (*E*)-1,3-pentadiene, 3-methyl-pentane, 2-methyl-1-propylene, (*E*)-2-pentene
Edible + inedible	4	3-methylfuran, 2,2,4,6,6-pentamethylheptane, 2,2,4,4-tetramethyloctane, 1-pentene
Preferred only	21	(1S)-1,7,7-trimethyldicycloheptyl-2-one, camphene, 1,7,7-trimethyl-tricycloheptane, sec-butyl ester-cyanic acid, (1R)-2,6, 6-trimethyldicycloheptyl-2-ene, α-pinene, 3-methyl-2-butanol, methylsilane, 2-methyl-1-butene, *N*,*N*,*N*’,*N*’-tetramethylenediamine, (1S)-6,6-dimethyl-2-methylene-dicycloheptane, 1-benzyl-3-amino-4-cyano-3-pyrroline, (S)-2-methylbutan-1-ol, (R)-(–) -2-pentanol, 2-ethyl-furan, carbonyl sulfur, 4-tetramethyl-1,3-cyclobutanediol, ethanol, hexamethyldisiloxane, *N*-dimethylaminomethyl tert-butyl isopropyl phosphine, 1,3-propanediol
Edible only	7	2,4-dimethyl-1,3-pentadiene, (*Z*)-1,3-pentadiene, 2,2-dimethyldecane, 1-vinyl-4-methoxy-benzene, 2-butenyl-hydrazine, propyl cyclopropane, 2-methyl-3-butyl-2-ol
Inedible only	20	2-pentyl-furan, dichloromethane, cyclohexane, 3-ethyl-2, 2-dimethyl-pentane, dimethylsulfide, 2-methoxy-ethanol, 1-undecene, 1-[(2-methyl-2-allyl)oxy ]-butane, tri-(trimethylmethysilyl) borate, fluoropropylene, 2-methyl-1,3-butadiene, 2-fluoro-propylene, 3,4-dimethyl-2-hexanone, (*Z*)-2-pentene, 1-methyl-butyl-oxirane, 3-isopropyl-5,5-dimethyl-cyclopentene, formic acid, 3-dioxazole-2-one, *N*-propyl-3,4-methylene dioxyamphetamine, octadecylcyclotetrasiloxane

**Table 3 plants-14-03856-t003:** Identification of metabolites by HPLC-MS and sources of information.

Ionization Mode	Number of Peaks	Total Number of Identified Metabolites	Number Identified in Public Databases	Number Identified in KEGG
+	16,208	1238	1011	343
−	18,516	1157	1070	241

**Table 4 plants-14-03856-t004:** Metabolites specific to the preferred, edible and inedible groups of bamboo shoots.

Bamboo Group	Unique Cations	Unique Anions
Preferred	*Fusarium* chromone, darfpristin, 1-(9H pyridine[3,4-β]indole-1-yl)-1,4 -butanediol, fluorescein D2, zanthobisquinone, Hv-NCC-1, β-humulene	pollen B, acetylcoumarol, holdatin A glucoside, cordosion, scleroporphyrin, jubanin A, 13′-carboxyl γ-tocopherol, 22α-hydroxyerythritol, kaempferol 3-sophoricoside 7-glucuronide, (R)-1-O-[β-d-methylfuranyl-(1→2)-β-d-glucopyranoside]-1,3-octylglycol, [3,4-dihydroxy-4-(1-oxo-1H-isochromen-3-yl)butoxy] sulfonic acid, {[3,4-dihydroxy-4-(1-oxo-1H-isochromen-3-yl) butan-2-yl]oxy} sulfonic acid, armexifolin, methylnorlichexant
Edible	aflatoxin G2a, 6-methoxygalacturonic acid 7-glucuronic acid, dry acid	7,7′- dihydroxy-6,8′-dicoumarin, 6′′-O-acetylglycine, dihydroxyanthraquinone, PA (16:1 (9Z)/18:3 (6Z, 9Z, 12Z))
Inedible	spermidine, gemaklenone, sphingosine (1+), 4-α-methyl-5-α-cholesterol-7-en-3-one, dynorphin A (6-8), 1-(5-methyl-3-pyridyl)-1-decanone, 3-hydroxyundecylcarnitine, phenethylacetate	cycloharringtonin C, TG, 4-β-hydroxymethyl-4-α-methyl-5-α-choleste-7-en-3-β-ol, cysteine, 2-(3,5-dihydroxy-4-methoxyphenyl)- 4H-chrome-4-one

**Table 5 plants-14-03856-t005:** Detection of cations and anions differing in abundance when assessed by pairwise comparison. Numbers in parentheses refer to identified ions, which are represented in Figure 8.

Ionization Mode	Total Ion Number	Preferred vs. Edible Groups	Preferred vs. Inedible Groups	Inedible vs. Edible Groups
+	1214 (152)	718 (97)	839 (116)	243 (18)
−	1644 (168)	990 (85)	1066 (124)	406 (50)
Total	2858 (320)	1708 (182)	1905 (240)	649 (68)

## Data Availability

The original contributions presented in this study are included in the article/Appendix A. Further inquiries can be directed to the corresponding authors.

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
