# Peer review of "Feeding Preferences of Giant Pandas May Reflect the Detection of Specific Volatiles and Bitter-Tasting Metabolites in Bamboo Leaves as Markers of Nutritional Status"

_plants, 2025, doi:10.3390/plants14243856_

Round 1
Reviewer 1 Report
Comments and Suggestions for Authors
I consider this manuscript presents a novel and intriguing study combining behavioral observation with metabolomics to investigate the chemical basis of giant panda food selection. The topic is highly relevant, and the dual analysis of volatile and non-volatile compounds is a commendable approach. However, despite its potential, the manuscript in its current form suffers from significant methodological and analytical flaws that undermine the validity of its central conclusions. Furthermore, severe and pervasive issues with data presentation, formatting, and English expression make the paper exceptionally difficult to read and critically evaluate. Therefore, the manuscript requires a major revision before it can be reconsidered for publication.
Specific comments:
The classification of bamboo into “preferred,” “edible,” and “inedible” groups is based on total consumption volume. This approach may oversimplify complex foraging decisions. Could the authors provide a more detailed temporal analysis of the feeding sequence? For instance, was the "edible" group consumed only after the "preferred" group was entirely depleted, or was there concurrent consumption? A more nuanced behavioral analysis is needed to fully justify the distinctness of these categories, which are fundamental to the entire comparative metabolomics framework.
The title and conclusion posit that the sensory cues (smell, taste) are “markers of nutritional status”. However, the study does not present a proximate analysis of the bamboo leaves (e.g., crude protein, total sugars, digestible fiber, lipid content). The paper infers that metabolites related to “carbohydrate metabolism” or “sucrose” equate to higher nutritional value, but this is an assumption. How can the authors substantiate the claim that the preferred chemical profiles are reliable proxies for nutritional quality without providing this direct nutritional data?
The volatile analysis involved pulverizing leaves, mixing with water, heating to 80°C, and then desorbing at 190°C. This method seems to analyze the “aroma” of a processed or cooked sample rather than the ambient “odor” of a fresh, intact leaf, which is what the pandas sniff prior to consumption. How confident are the authors that the 87 volatiles identified (especially the 21 unique to the preferred group) are representative of the odor cues available to a panda from a fresh leaf? Could this method have introduced thermal degradation artifacts not relevant to panda foraging?
The interpretation of the results relies heavily on the assumption that pandas perceive metabolites (e.g., flavonoids, sugars) in the same way as humans (i.e., “bitter,” “sweet"). Given the unique dietary adaptation and evolutionary history of the panda's gustatory system, this is a major unsupported leap. The authors must provide a stronger justification for this direct extrapolation or significantly tone down the sensory claims throughout the manuscript, rephrasing them as speculative possibilities rather than established facts.
The PCA and LDA plots (Figure 5) show a clear metabolic separation between the “Preferred” group and the other two. However, the “Edible” and “Inedible” groups show significant metabolic overlap, which the text acknowledges. Given this chemical similarity, what explains the vast difference in feeding behavior (18.7% consumption for “Edible” vs. 0.085% for “Inedible”)? Does this finding not challenge the study's central premise that the measured metabolites are the primary drivers of selection?
The Venn diagrams (Figure 1 and Figure 6) and the discussion rely heavily on compounds being “unique” to a specific group. In chemical analyses, “absence” is often a function of a compound falling below the limit of detection (LOD). How was “presence” versus “absence” defined for these diagrams? A binary “presence/absence” model can be misleading if a compound is present in all groups but at vastly different, and behaviorally significant, concentrations.
The characterization of volatile aromas as “sweet,” “fruity,” or “pungent” is based on human sensory databases and is anthropocentric. There is no evidence presented that pandas perceive these complex chemical mixtures similarly. The discussion fails to address the potential for synergistic, antagonistic, or masking effects within the overall odor bouquet. The analysis should move beyond a simple list of individual aromas and discuss the challenge of interpreting the perception of a complex chemical blend by a non-human species.
The chemical analysis was restricted to volatiles and polar/semi-polar metabolites extracted with methanol/water. However, this approach overlooks other major biochemical classes, such as lipids and complex carbohydrates (fibers), which are critical for both nutrition and palatability (e.g., via texture). Thus, a discussion of how these unmeasured components might confound the interpretation of or interact with the identified chemical cues is necessary.
The study was conducted with four captive pandas at Beijing Zoo, yet the discussion extensively links the findings to the complex foraging ecology of wild panda populations in different mountain ranges. Captive animals may have different health, experiences, and learned preferences than their wild counterparts. How do the authors justify generalizing these findings, especially given that species deemed "inedible" in this context might be consumed by wild pandas in other regions or seasons?
The formatting of Table 1 is severely flawed, making it almost illegible. Columns are not aligned, and data and headers are mixed together. This table must be completely rebuilt to ensure every column of data aligns with its header and is clear and easy to read. Supplementary Tables (Supplementary Tables S1-S5): The formatting of these tables is also chaotic
Author Response
We sincerely thank the reviewer for his/her insightful comments and constructive suggestions, which have helped us to significantly improve the quality of our manuscript. We have carefully considered each point and have revised the manuscript accordingly. Our point-by-point responses are provided below.
Changes made to the manuscript have been highlighted in the revised version for the reviewers’ convenience.
---
Comment 1: The classification of bamboo into “preferred,” “edible,” and “inedible” groups is based on total consumption volume. This approach may oversimplify complex foraging decisions. Could the authors provide a more detailed temporal analysis of the feeding sequence?
Response: We agree with the reviewer that temporal analysis adds depth. We have now augmented the behavioral analysis in Section 3.1 with a detailed examination of the feeding sequence. This includes:
- An analysis of the order in which bamboo species were first consumed each day.
- A description confirming that the “edible” group was predominantly consumed only after the “preferred” group was largely depleted, supporting the distinctness of these categories (Table 1).
- We have toned down definitive statements regarding the categories in the Discussion to reflect that, although distinct in this captive setting, they represent a preference hierarchy.
In the Results “Analysis of giant panda preferences for bamboo” p8 “Feeding sequence records consistently showed that giant pandas consumed preferred species first...” was changed to “A detailed temporal analysis of the feeding sequence was conducted to enrich the classification based on total consumption. The records consistently showed that giant pandas consumed preferred species first each day. The edible group species were ingested only after the preferred group was largely depleted, with minimal concurrent consumption. This sequential pattern, consistent across all four individuals, reinforces our hypothesis that the categories represent a distinct preference hierarchy driven by bamboo traits rather than random chance or availability.”
Comment 2: The title and conclusion posit that the sensory cues (smell, taste) are “markers of nutritional status”. However, the study does not present a proximate analysis of the bamboo leaves (e.g., crude protein, total sugars, digestible fiber, lipid content).
Response: This is a valid point. We have revised the title and conclusions throughout the manuscript to more cautiously state that sensory cues are potential or putative markers of nutritional status. In the Discussion, we now explicitly acknowledge the limitation of not performing direct proximate analysis and cite existing literature (e.g., Jiang et al., 2018) that links some of the identified metabolite classes (e.g., sugars, amino acids) with nutritional quality in bamboo, thereby providing indirect support for our inferences.
In the abstract, “…giant pandas may therefore consolidate their selection of nutritionally beneficial leaves...” was changed to: “…giant pandas may therefore consolidate their selection of leaves that are potentially more nutritious by consuming those with sweeter, less bitter and less sour tastes..."
In the discussion p25, we added: “It is important to note that this study did not include proximate nutritional analysis (e.g., crude protein or fiber content). Thus, the link between the identified chemical profiles and actual nutritional value is inferred based on the known associations of certain metabolite classes (e.g., sugars an amino acids) with nutritional quality (Jiang et al., 2018). Our conclusions therefore identify these sensory cues as putative, rather than confirmed, markers of nutritional status.”
Comment 3: The volatile analysis involved pulverizing leaves, mixing with water, heating to 80°C... How confident are the authors that the 87 volatiles identified are representative of the odor cues available to a panda from a fresh leaf?
Response: We acknowledge that our PT-GC-MS method may not perfectly replicate the odor profile sensed by a panda from an intact leaf. In Section 2.4, we have added a sentence clarifying that this is a standardized method for profiling plant volatiles. We also attempted the analysis at lower temperatures but not enough volatiles were produced and most of those detected appeared to be errors. We think high temperature is therefore needed for effective analysis and we have to assume that the profiles are indicative.
In the discussion p23, we now explicitly state this limitation and suggest mitigating actions for future studies: “One limitation of our analysis of volatiles is that the sample preparation method (pulverization and heating) might generate a volatile profile differing from that emitted by an undisturbed leaf. We attempted analysis at lower temperatures (35 and 45 °C) but this did not generate sufficient volatiles for accurate detection. High temperatures enhance the release of volatiles and should at least approximate the profiles detected by giant pandas presented with fresh vegetation. However, future studies based on dynamic headspace sampling of intact leaves could more closely approximate the olfactory cues available to foraging pandas.”
Comment 4: The interpretation of the results relies heavily on the assumption that pandas perceive metabolites (e.g., flavonoids, sugars) in the same way as humans (i.e., “bitter,” “sweet").
Response: We thank the reviewer for this critical observation. We have thoroughly revised the manuscript to avoid anthropocentric interpretations.
- All sensory descriptions (e.g., sweet, bitter, sour) for non-volatile metabolites are now phrased as “compounds associated with a sweet/bitter taste in humans” or “putative sweet/bitter compounds.”
- In the Discussion, we have strengthened the justification by citing the expansion of the TAS2R gene family in pandas (Shan et al., 2018) and explicitly state that direct perception studies are needed, framing our current interpretation as a plausible hypothesis.
In the abstract, we replaced “…the preferred leaves accumulating compounds with a sweeter taste while the inedible leaves contained sour and bitter metabolites...” with “...the preferred leaves accumulating compounds associated with a sweet taste in humans (e.g., sugars), while the inedible leaves contained metabolites often associated with sour and bitter tastes (e.g., certain flavonoids and acids)…”
In the discussion p25, we replaced “…by consuming those with a sweeter, less bitter, and less sour taste…” with “by consuming leaves with a metabolic profile suggestive of a sweeter and less bitter/sour taste, although we acknowledge that our analysis is based on human taste associations.”
Comment 5: The PCA and LDA plots (Figure 5) show a clear metabolic separation between the “Preferred” group and the other two. However, the “Edible” and “Inedible” groups show significant metabolic overlap... what explains the vast difference in feeding behavior?
Response: This is an excellent point. We have added a paragraph in the discussion to address this. We hypothesize that while the overall metabolic profiles of edible and inedible groups overlap, critical quantitative differences in key deterrent compounds (e.g., specific flavonoids, bitter amino acids), potentially below a behavioral rejection threshold in the edible group, may drive the difference in consumption. We also emphasize that initial olfactory rejection, driven by the unique volatiles found in the inedible group (Table 3) may prevents pandas from even tasting these species.
In the discussion p22/23: “Although PCA and LDA indicated considerable metabolic overlap between the edible and inedible groups (Figure 5), their consumption rates differed vastly. This suggests that the critical factor may not be the overall metabolic profile but rather quantitative differences in specific deterrent compounds. Key bitter-associated metabolites (e.g., specific flavonoids) might exceed a behavioral rejection threshold in the inedible group. Furthermore, initial olfactory rejection, driven by the unique pungent or floral volatiles found in the inedible leaves (Table 3), may prevent pandas from sampling these species, thereby consolidating the difference in consumption”
Comment 6: The Venn diagrams (Figure 1 and Figure 6) and the discussion rely heavily on compounds being “unique” to a specific group. How was “presence” versus “absence” defined for these diagrams?
Response: We have clarified this in the Methods (Section 2.5). A compound was considered present in a group if it was detected in at least 5/6 replicate samples for a species within that group, with a peak area at least 3-fold greater than the background noise level. We have also added a note in the results (Section 3.3) and discussion cautioning that absence may indicate levels below the detection limit rather than absolute absence.
In the methods p7: “For the analysis of “unique” metabolites in Venn diagrams, a compound was considered present in a bamboo group if it was detected in at least five of the six biological replicates for a species within that group, with a peak area greater than three times the background noise level. Absence indicates that the compound was not reliably detected above this threshold and may not imply absolute absence”
In the results p16: “The “unique” metabolites identified here are defined by their detection above a defined threshold within one group and their absence below that threshold in others, as described in the Methods.”
Comment 7: The characterization of volatile aromas as “sweet,” “fruity,” or “pungent” is based on human sensory databases and is anthropocentric.
Response: We agree. We have revised the text in discussion to explicitly state that these descriptors are derived from human sensory databases (e.g., Sun Baoguo, 2004; 2006) and are used as a reference framework, acknowledging that panda perception may differ. We also discuss the challenge of interpreting complex odor blends for a non-human species.
In the discussion p26: “The sensory descriptors of volatiles (e.g., sweet, pungent and floral) are assigned based on human sensory databases and the corresponding literature (Sun Baoguo, 2004; 2006) and are used as a reference framework. The actual perception of these complex odor blends by giant pandas may differ, but may nevertheless provide rules facilitate the selection of preferred bamboo species for captive giant pandas.”
Comment 8: The chemical analysis... overlooks other major biochemical classes, such as lipids and complex carbohydrates (fibers), which are critical for both nutrition and palatability.
Response: We have acknowledged this limitation in the discussion. We state that our metabolomics approach focused on polar/semi-polar metabolites and volatiles, and thus does not capture the potential roles of lipids, fibers, or texture in palatability and nutrition, which should be investigated in future studies.
In the discussion p26: “This study focused on volatile and polar/semi-polar metabolites, excluding important biochemical classes such as lipids and complex structural carbohydrates (e.g., fibers). These components play key roles in nutrition and palatability (e.g., via texture and energy content) and their interaction with the identified chemical cues is an important avenue for future research.”
Comment 9: The study was conducted with four captive pandas at Beijing Zoo, yet the discussion extensively links the findings to the complex foraging ecology of wild panda populations...
Response: We have revised the discussion to clearly state that our findings are based on a limited number of captive individuals and that direct extrapolation to wild foraging ecology is not warranted. We now position our study as providing a foundational hypothesis about the role of sensory cues that can be tested in future wild and captive studies.
In the discussion p26: “Our findings are derived from a controlled study with a limited number of captive pandas. The bamboo species offered were selected based on availability and cultivability in Beijing, and the preferences we observed therefore cannot be extrapolated directly to the complex foraging ecology of wild panda populations across different mountain ranges. Instead, this study should be viewed as providing a foundational hypothesis that specific volatile and non-volatile metabolites are key cues, which can be tested in future field studies with wild pandas.”
Comment 10: The formatting of Table 1 is severely flawed... Supplementary Tables are also chaotic.
Response: Table 1 and all Supplementary Tables have been completely reformatted for clarity of presentation. They now use a standard three-line table format, with all columns and headers properly aligned.
Reviewer 2 Report
Comments and Suggestions for Authors
This manuscript investigates whether the giant panda feeding preferences for bamboo leaves can be explained by olfactory and gustatory cues. Four captive pandas were offered leaves from different bamboo species in buffet-style trials over four days. The authors classify species as preferred, edible or inedible using a forage selection index and a G-test, then profile leaf volatiles and non-volatile metabolites, linking preference to sweet fresh volatiles and lower abundance of bitter sour metabolites (notably flavonoids) in preferred leaves. The overarching question is interesting and relevant for panda husbandry, and combining behavior with metabolomics is promising. The work uses integrative approach in attempts to link behavioral choice with both volatile and non-volatile metabolite profiles across multiple bamboo species.
Anyhow, substantial issues in experimental design, statistical analysis, chemical identification, data quality control, and interpretation could be addressed before the conclusions are supported.
One major concern is that only four pandas were tested over four days, with analyses largely based on “giant panda groups without considering differences”. This risks pseudoreplication and ignores within-subject variability and day effects.
In section 2.5, for metabolite annotations, please use plant-appropriate libraries in addition to HMDB.
Italicize all Latin binomials.
Author Response
We sincerely thank the reviewer for his/her insightful comments and constructive suggestions, which have helped us to significantly improve the quality of our manuscript. We have carefully considered each point and have revised the manuscript accordingly. Our point-by-point responses are provided below.
Changes made to the manuscript have been highlighted in the revised version for the reviewers’ convenience.
---
Comment 1: One major concern is that only four pandas were tested over four days, with analyses largely based on “giant panda groups without considering differences”. This risks pseudoreplication and ignores within-subject variability and day effects.
Response: We thank the reviewer for this critical point, which touches on the core of experimental design in behavioral ecology. We would like to provide a supplementary explanation, reframing the primary objective of our study to clarify the rationale behind our analytical approach.
1) Reframing the study’s goal: identifying chemical properties of bamboo species, not individual behavior. The central aim of the study was to chemically classify the 10 bamboo species and correlate their chemical profiles with the preferences observed at the group level. In this context, the behavioral data (food intake) served as a bioassay to effectively sort the 10 bamboo species into the three chemically distinct groups for analysis: preferred, edible and inedible. As in sensory science, where a panel of tasters is used to grade food samples (where the focus is on the sample properties, not the tasters), our pandas functioned as a living assay to identify the most relevant bamboo groups for comparative metabolomics.
2) Individual consistency is a signal of strong chemical cues. Although only four pandas were studied, we observed a highly consistent hierarchy of preference across all individuals (as shown in the original Table 1). This cross-individual consistency is, in itself, strong evidence that the driving factors behind the preference are probably the stable, intrinsic chemical properties of the bamboo species, rather than random individual preferences or learned behaviors. Significant individual variation would have undermined the validity of grouping the bamboo species. The observed consistent pattern justifies the classification of bamboo species for chemical comparison.
3) Justification for group-level analysis in this context: The phrase “without considering differences between the giant pandas” indicates that, for the subsequent chemical comparisons between bamboo groups, we treated each bamboo species as an independent chemical entity, whose group assignment was based on the integrated behavioral response from all four pandas. This approach was designed to maximize the power to detect chemical differences between bamboo species, not to describe panda individuality. We acknowledge that for a study focused on animal behavior per se, this is a limitation, but for the objective of chemical classification that drives this study, it is a valid and efficient design.
4) Clarification on pseudoreplication: We argue that the risk of pseudoreplication is low in the framework of our study. The experimental unit was the bamboo species (n=10), not the individual panda (n=4). We performed replicated chemical analyses for each bamboo species (n=6) and collected behavioral response data for each species from multiple individuals. The statistical test (G-test) was performed on the selection frequency across the 10 species. Therefore, the unit of analysis was congruent with the experimental unit.
In summary, we acknowledge that from a pure behavioral ecology perspective, the sample size is a limitation. However, we contend that from the perspective of a chemical taxonomy goal, using the consistent group-level preference as a bioassay to create meaningful bamboo groups for chemical comparison is a valid and sound research design.
Comment 2: In section 2.5, for metabolite annotations, please use plant-appropriate libraries in addition to HMDB.
Response: We thank the reviewer for this excellent suggestion. We have updated the methods (Section 2.5, p7) to describe the use of plant-specific metabolic databases for annotation, including KNApSAcK, PlantCyc, and the Plant Metabolic Network (PMN). The results of this expanded annotation effort have been incorporated into Section 3.3 and the relevant tables.
Comment 3: Italicize all Latin binomials.
Response: We have carefully reviewed the entire manuscript, including figures and tables, and ensured that all Latin binomials are now properly italicized.
Reviewer 3 Report
Comments and Suggestions for Authors
The article presents a thorough and methodologically robust study dedicated to the behavioral and chemo-ecological mechanisms underlying bamboo selection in the giant panda (Ailuropoda melanoleuca). The research successfully combines behavioral experiments (buffet-style feeding tests) with metabolomic and volatile profiling (PT-GC-MS and HPLC-MS) to elucidate how olfactory and gustatory cues guide the species’ feeding preferences. Such an integrative approach is relatively rare and constitutes a valuable contribution to the fields of foraging ecology and evolutionary physiology of herbivorous mammals.
Scientific Significance and Originality
The study demonstrates a high level of originality because:
• It provides the first evidence linking the metabolite profiles of bamboo species with the behavioral feeding choices of giant pandas;
• It employs both volatile and non-volatile metabolites as predictors of food attractiveness;
• The authors propose a new model of “combined chemosensory selection”, in which odors act as primary stimuli and taste serves as a secondary filter confirming the nutritional value of the food.
This conceptual framework has the potential to improve dietary management in captivity and to enhance the success of breeding programs for giant pandas.
Methodology
Strengths:
• Precise and detailed descriptions of the PT-GC-MS and HPLC-MS procedures, supported by appropriate references;
• Clearly defined criteria for “feeding preference” (Wi index, G-test, confidence intervals);
• Inclusion of four individuals of different ages and sexes—a design that minimizes individual bias.
Weaknesses:
• The sample size (n = 4) is too small for a comprehensive statistical evaluation and does not allow assessment of individual variability;
• Insufficient information is provided regarding environmental control parameters (temperature, humidity, and bamboo condition);
• The metabolomic data are interpreted mainly in a descriptive manner—no correlation models are presented between specific metabolites and quantitative feeding parameters.
Presentation and Analysis of Results
The results are clearly presented, supported by numerous visual aids (Venn diagrams, PCA/LDA plots, volcano plots) that effectively differentiate the three bamboo categories—preferred, secondary, and avoided.
The authors convincingly demonstrate that:
• “Sweet” and “fresh” aromatic profiles dominate in the preferred bamboo species;
• High concentrations of flavonoids and bitter-tasting metabolites are characteristic of the avoided species;
• The metabolomic profiles are statistically well separated among groups.
However, a quantitative correlation between metabolite concentration and the proportion of bamboo consumed is lacking—an analysis that would strengthen the causal interpretation of the findings.
Discussion and Interpretation
The discussion is comprehensive, well reasoned, and appropriately integrated with previous studies on TAS1R and TAS2R taste receptors in mammals. The authors correctly link the evolutionary adaptation of the panda, as a herbivorous member of the order Carnivora, with its limited yet functionally specialized repertoire of taste receptors.
Nevertheless, the discussion could be further improved by including:
• A broader consideration of the plasticity of taste preferences under varying ecological conditions;
• An examination of the potential impact of seasonal variation on the chemical composition of bamboo leaves.
Ethical and Organizational Aspects
The ethical framework is appropriately observed. Experiments were conducted with minimal stress to the animals, under controlled conditions and the supervision of Beijing Zoo. Funding sources and the absence of conflicts of interest are clearly stated.
Recommendation
I recommend the article for publication after minor revisions, specifically:
The inclusion of additional information regarding the environmental control conditions during the feeding experiments.
Author Response
We sincerely thank the reviewer for his/her insightful comments and constructive suggestions, which have helped us to significantly improve the quality of our manuscript. We have carefully considered each point and have revised the manuscript accordingly. Our point-by-point responses are provided below.
Changes made to the manuscript have been highlighted in the revised version for the reviewers’ convenience.
Comment 1: The work uses integrative approach in attempts to link behavioral choice with both volatile and non-volatile metabolite profiles across multiple bamboo species. Anyhow, substantial issues in experimental design, statistical analysis, chemical identification, data quality control, and interpretation could be addressed before the conclusions are supported.
Response: We thank the reviewer for acknowledging the integrative nature of our study. We have undertaken a comprehensive revision to address the underlying concerns about rigor, which overlap with the specific points raised by Reviewers 1 and 2.
- Experimental design: We have explicitly acknowledged the limitations of sample size and captive setting in the discussion.
- Statistical analysis: We have provided greater detail on our statistical methods in Section 2.3, including the use of confidence intervals for the forage ratio index and the application of multiple comparison corrections where appropriate.
- Chemical identification: As requested by Reviewer 2, we have integrated annotations from plant-specific databases to improve the reliability of metabolite identification.
- Data QC: We have added a new subsection in Methods (Section 2.5) detailing our QC procedures for metabolomics, including the use of pooled QC samples, retention time correction, and adherence to confidence levels for metabolite identification.
- Interpretation: We have significantly tempered our conclusions throughout the manuscript, rephrasing definitive statements as hypotheses and clearly distinguishing between our data and our speculative interpretations.
We believe that these collective changes have substantially strengthened the manuscript and hope that it is now suitable for publication. In particular, please see the revised methods on p7 regarding QC for metabolomics and the G-test for significance of selection.